# Production of Demineralized Antibacterial, Antifungal and Antioxidant Peptides from Bovine Hemoglobin Using an Optimized Multiple-Step System: Electrodialysis with Bipolar Membrane

**DOI:** 10.3390/membranes12050512

**Published:** 2022-05-11

**Authors:** Mira Abou-Diab, Jacinthe Thibodeau, Ismail Fliss, Pascal Dhulster, Laurent Bazinet, Naima Nedjar

**Affiliations:** 1Department of Food Science, Université Laval, Québec, QC G1V 0A6, Canada; mira.abou-diab.1@ulaval.ca (M.A.-D.); jacinthe.thibodeau.1@ulaval.ca (J.T.); ismail.fliss@fsaa.ulaval.ca (I.F.); 2Laboratoire de Transformation Alimentaire et Procédés ÉlectroMembranaires (LTAPEM, Laboratory of Food Processing and Electro-Membrane Processes), Université Laval, Québec, QC G1V 0A6, Canada; 3Institute of Nutrition and Functional Foods (INAF), Université Laval, Québec, QC G1V 0A6, Canada; 4UMR Transfrontalière BioEcoAgro N°1158, Université Lille, INRAE, Université Liège, UPJV, YNCREA, Université Artois, Université Littoral Côte d’Opale, ICV—Institut Charles Viollette, F-59000 Lille, France; pascal.dhulster@univ-lille.fr

**Keywords:** electrodialysis with bipolar membrane, bovine hemoglobin, demineralized hydrolysates, discolored-demineralized hydrolysates, antibacterial activity, antifungal activity, antioxidant activity

## Abstract

Numerous studies have shown that bovine hemoglobin, a protein from slaughterhouse waste, has important biological potential after conventional enzymatic hydrolysis. However, the active peptides could not be considered pure since they contained mineral salts. Therefore, an optimized multi-step process of electrodialysis with bipolar membranes (EDBM) was carried out to produce discolored and demineralized peptides without the addition of chemical agents. The aim of this study was to test the antibacterial, antifungal and antioxidant activities of discolored and demineralized bovine hemoglobin hydrolysates recovered by EDBM and to compare them with raw and discolored hydrolysates derived from conventional hydrolysis. The results demonstrate that discolored–demineralized hydrolysates recovered from EDBM had significant antimicrobial activity against many bacterial (gram-positive and gram-negative) and fungal (molds and yeast) strains. Concerning antibacterial activity, lower MIC values for hydrolysates were registered against *Staphylococcus aureus*, *Kocuria rhizophila* and *Listeria monocytogenes*. For antifungal activity, lower MIC values for hydrolysates were registered against *Paecilomyces* spp., *Rhodotorula mucilaginosa* and *Mucor racemosus*. Hemoglobin hydrolysates showed fungicidal mechanisms towards these fungal strains since the MFC/MIC ratio was ≤4. The hydrolysates also showed a potent antioxidant effect in four different antioxidant tests. Consequently, they can be considered promising natural, low-salt food preservatives. To the best of our knowledge, no previous studies have identified the biological properties of discolored and demineralized bovine hemoglobin hydrolysates.

## 1. Introduction

According to the World Health Organization (WHO), food contamination is a serious problem affecting many consumers across the world. Generally, food is considered contaminated when its flavor, texture and/or appearance change, making it unhealthy for consumption. The major causes of deleterious quality changes in food are due to microbial, chemical and physical contamination or oxidation [1].

Microbial contamination happens when a product has been contaminated by microorganisms, including bacteria, fungi, yeasts, protozoa and even viruses [2]. Microbial contamination of food has become the biggest issue source of concern for the industry. Pathogenic microbes can cause several diseases, such as botulism, food poisoning and certain other enteric infections by producing harmful toxic metabolites [3]. The most common food pathogenic bacteria are *Salmonella* spp., *Listeria monocytogenes*, *Staphylococcus aureus*, *Escherichia coli*, *Clostridium botulinum*, *Shigella* spp., *Pseudomonas* spp. and *Campylobacter jejuni* [1]. Fungi are also major food contaminants, the most common fungal species, including molds and yeasts, found as food contaminant are *Aspergillus niger*, *Eurotium rubrum*, *Fusarium* sp., *Paecilomyces* spp., *Penicillium citrinum*, *Penicillium crustosum*, *Rhodotorula mucilaginosa*, *Candida intermedia* and *Saccharomyces boulardii* [4]. In addition, oxidation of biomolecules has many undesirable impacts on organoleptic quality in food. Furthermore, several diseases can appear due to oxidative stress, such as cancer, arteriosclerosis, cardiovascular diseases, diabetes mellitus, neurological disorders and Alzheimer, since it can modify proteins, DNA and small cellular molecules [5,6].

To control microorganisms and oxidation reactions, preservatives are regularly added to food. The most commonly used antimicrobial and antioxidant compounds for the preservation of foods are: organic acids (acetic acid, benzoic acid, sorbic acid, etc.), nitrites and nitrates, sulfur dioxide and sulfites, ethylene oxide and propylene oxide, sugars, salts, alcohol, antibiotics, etc. [1]. However, some of these are suspected to induce pathological and toxic effects. Moreover, microorganisms are becoming resistant to antibiotics and preservatives [7]. For this reason, and to respond to growing consumer demand for healthy and quality foods, interest in producing natural preservatives has risen in the past decades. Bio-preservatives are produced from natural sources or formed in food and can prevent or delay spoilage related to chemical or biological deterioration and consequently extend shelf life.

In this context, and in the framework of circular economy, bovine blood from slaughterhouse waste was demonstrated to allow the production of natural antimicrobial peptides [8,9]. Moreover, the α137-141 peptide derived from bovine cruor waste has been demonstrated to be a promising natural substitute for the synthetic additives widely used to protect meat during storage and distribution by reducing lipid oxidation by 60% and inhibiting growth of total viable colonies, yeasts and molds [10]. Furthermore, the biological properties (antibacterial, antifungal and antioxidant) of bovine hemoglobin hydrolysates from bovine blood slaughterhouse waste, produced using an innovative and green process: electrodialysis with bipolar membrane (EDBM), have been proven [11]. However, further experiments were needed to: (1) settle the issue of membrane fouling that appeared during the production of hydrolysates as explained by Abou-Diab et al. [12]; (2) remove, by discoloration, the heme contained in the hydrolysates, which is responsible for the red color of blood and makes product valorization difficult due to iron oxidation; and (3) reduce the final mineral salt concentration to be able to meet consumer demands.

Therefore, a new optimized study performing a multiple-step EDBM process was carried out recently by Abou-Diab et al. [13] to produce demineralized and discolored–demineralized bovine hemoglobin hydrolysates without the addition of mineral salts. The optimized EDBM configuration consisted of three-compartment combining homopolar membranes with bipolar membranes, whose membranes generate H^+^ and OH^−^ ions from water dissociation under an electrical field. Such an optimized EDBM configuration allowed simultaneous: enzymatic hydrolysis, inactivation of reaction, discoloration of the hydrolysate (to suppress fouling observed on the anionic membrane due to heme precipitation) and a final demineralization of 85% of the discolored hydrolysate. This integrated process has made possible the transformation of a by-product from slaughterhouses into a discolored and demineralized product (EDBM-DH) containing bioactive peptides. However, in this previous study, the authors focused on the technological part of the process to produce bioactive peptides from bovine hemoglobin without membrane fouling, the major concern in membrane processes application, and with a low salt concentration, but they did not test the final biological activities of the hydrolysates obtained with such optimized technology. Indeed, the biological bioactivities of the hydrolysates must be preserved to be able to use this new EDBM process in a circular economy.

In this context, the objective of the present study was to identify the antibacterial, antifungal and antioxidant activities of demineralized and discolored-demineralized bovine hemoglobin hydrolysates produced by EDBM in the previous study. The results will be compared to raw and discolored hydrolysates produced using conventional, multi-chemical-based hydrolysis (CH), as reported in [13]. For this purpose, the antibacterial and antifungal potentials were investigated using agar diffusion and minimum inhibitory concentration against multiple pathogenic and non-pathogenic strains commonly known as food contaminants. In addition, the antioxidant potential was investigated using four tests based on different mechanisms, including β-carotene bleaching inhibition activity, DPPH radical scavenging activity assay, ABTS radical scavenging activity assay and evaluation of total antioxidant capacity. To the best of our knowledge, there is no information concerning the biological properties of a demineralized bovine hemoglobin hydrolysate.

## 2. Materials and Methods

### 2.1. Materials

Hemoglobin (H2625, purified powder from bovine blood), pepsin (P6887, lyophilized powder from porcine gastric mucosa), reagents (KCl, Na_2_SO_4_, KOH and HCl), chemicals (1,1-diphenyl-2-picrylhydrazyl (DPPH), butylated hydroxytoluene (BHT), 2,20-azino-bis 3-ethylbenzothiazoline-6-sulphonic acid (ABTS), 6-hydroxy-2,5,7,8-tetramethylchroman-2-carboxylic acid (Trolox), β-carotene and linoleic acid), neokyotorphin (TSKYR, α137-141), bacterial strains (*Listeria monocytogenes* (ATCC 19112), *Staphylococcus aureus* (ATCC 13709), *Micrococcus luteus* (ATCC 9341), *Escherichia coli* (ATCC 8733), *Salmonella newport* (ATCC 6962) and *Kocuria rhizophila* (CIP 53.45)) and fungal strains (*Aspergillus niger* (3071-13), *Paecilomyces* spp. (5332-9a), *Mucor racemosus* (LMA-722), *Penicillium crustosum* (27,159) and *Rhodotorula mucilaginosa* (27,173)) were purchased, supplied and isolated from the same companies as described by Abou-Diab et al. [11].

### 2.2. Production of Hemoglobin Hydrolysates Using EDBM

#### 2.2.1. Electrodialysis Cell

As described by Abou-Diab et al. [13], the electrodialysis (ED) experiments were performed using a MP type cell with an effective surface area of 100 cm^2^. In summary, the ED configuration consisted of one anion-exchange membrane (AEM), one monovalent cation perm-selective membrane (MCP) and two bipolar membranes (BM) purchased from Astom, Ameridia (Tokyo, Japan). To allow continuous recirculation, this arrangement defined 4 closed loops, where each was connected to a separate external tank (Figure 1).

#### 2.2.2. Preparation of Bovine Hemoglobin Demineralized Hydrolysate (EDBM-RH)

As described by Abou-Diab et al. [13], the production of demineralized hemoglobin hydrolysates requires many steps, including denaturation, enzymatic hydrolysis, pepsin inactivation and demineralization. However, before the steps were performed simultaneously, preliminary steps were carried out in EDBM without the addition of mineral salts to provide the solutions for all compartments.

To perform the hydrolysis, 15 g of purified bovine hemoglobin was solubilized in 100 mL of ultrapure water, followed by centrifugation, as described by Abou-Diab et al. [13]. Then, a 1% (*w*/*v*) hemoglobin solution was prepared from the recovered supernatant after determining the real concentration of Hb using Drabkin’s method. After that, the 1% Hb solution was transferred into the EDBM acidification compartment for denaturation by lowering the pH to 3 using the protons (H^+^) electro-generated at the cationic interface of the bipolar membrane after water dissociation under an electrical field. The enzymatic hydrolysis reaction was started by the addition of the pepsin at an enzyme/protein ratio of 1/11 (mole/mole) and performed at 32 °C and pH 3 for 3 h. Simultaneously, the reaction was stopped after 3 h in the basification compartment by increasing the pH to 9 using the hydroxide ions (OH^−^) produced at the anionic interface of the bipolar membrane. At the same time, the alkaline hydrolysate was demineralized in the diluate compartment due to the migration of anions and cations through the anionic and cationic membrane, respectively.

#### 2.2.3. Preparation of Bovine Hemoglobin Discolored-Demineralized Hydrolysate (EDBM-DH)

To perform discoloration, the pH of hemoglobin hydrolysates with inactive pepsin (pH = 9) recovered from the basification compartment was lowered to 4.5 using the electrogenerated HCl solution recovered from the acidification compartment as described by Abou-Diab et al. [13]. This step was done outside the ED system and followed by centrifugation at 6000× *g* for 30 min to separate the heme (non-protein part), responsible for the red color of the blood, from the globin (protein part) of the hemoglobin hydrolysate. Subsequently, the discolored hydrolysate was transferred into the diluate compartment for demineralization. In order to assess the repeatability, the experiments were performed in triplicate.

### 2.3. Production of Hemoglobin Hydrolysate Using the Conventional Method

A hemoglobin hydrolysate was also prepared using the conventional method as outlined in Abou-Diab et al. [13]. Acidification of hemoglobin was carried out by the addition of chemical HCl (2 M). After 3 h of hemoglobin hydrolysis with pepsin, chemical basification of hydrolysates was carried out by the addition of KOH (0.5 M), and therefore a raw hydrolysate was produced (CH-RH). Afterwards, chemical discoloration was performed on the alkaline hydrolysates by addition of HCl (2 M) followed by centrifugation, and therefore a discolored hydrolysate was recovered (CH-DH) without a subsequent demineralization step. Again, these experiments were performed in triplicate.

The biological properties were identified on raw and discolored hydrolysates recovered from EDBM and the conventional method as shown in Figure 2 to study the impact of demineralization and discoloration on biological activities.

### 2.4. Antibacterial Activity

#### 2.4.1. Agar Diffusion Method

Bacterial species used in this study were chosen because they are commonly known to be responsible for food alteration [14,15]. Five bacterial strains were used for the determination of antibacterial activity according to the method of Yaba Adje [16]: *Listeria monocytogenes* (ATCC 19112), *Staphylococcus aureus* (ATCC 13709), *Kocuria rhizophila* (CIP 53.45), *Micrococcus luteus* (ATCC 9341), *Escherichia coli* (ATCC 8733) and *Salmonella newport* (ATCC 6962). These bacterial species were kept at −20 °C in glycerol containing nutrient broth and were sub-cultured twice in Muller–Hinton broth (MH) at 30 °C under agitation (60 rpm) for *Micrococcus luteus* and at 37 °C for the other bacterial strains. The absorbance of the pre-cultures was determined after 24 h of incubation in nutrient broth at 620 nm, using MH as a blank. After a dilution series, seeding of the bacterial species at a concentration of ×10^6^ CFU/mL was carried out by the spot-on-lawn method on petri dishes containing 15 mL of MH agar previously casted and dried. A volume of 10 μL of the samples and the positive control were deposited on the surface of the agar after filtration (0.25 μm). Afterwards, plates were incubated 24 h at 30 °C for *Micrococcus luteus* and at 37 °C for the other bacterial strains. Antibacterial activity was measured as the diameter of the clear growth inhibition zone and recorded as diameter of inhibition in centimeters. Ampicillin (0.1 mg/mL) was used as positive control for all bacterial species and colistin (0.1 mg/mL) as positive control for *Escherichia coli*. This antibacterial test was performed in triplicate.

#### 2.4.2. Minimum Inhibitory Concentration (MIC)

The MIC assays were performed in sterile 96-well microplates (Costar 3799, Gillingham, UK) for which the response was read after 24 h incubation. Briefly, 100 μL of hemoglobin hydrolysate solution at a concentration of 80 mg/mL was added to 100 μL of MH in each well, serially diluted in half. At the end, 100 μL of the tested strain pre-diluted in MH (×10^5^ CFU/mL) was added to each well. After 24 h of incubation at 37 °C, the absorbance (600 nm) was measured using a microplate reader (Safas, model MP96 UV-Vis Spectrophotometer, Agilent Technologies, Santa Clara, CA, USA) to verify the inhibition of bacterial growth. The absorbance of the wells corresponding to decreasing concentrations of peptides was compared to that of the well of a positive control consisting of MH containing the bacteria and to that of the well of a negative control consisting of only MH. The MIC was defined as the lowest concentration of peptides that caused no visible increase of absorbance at 600 nm after incubation at 37 °C for 24 h. All measurements were performed in triplicate. All data are expressed as mean ± SD and are the mean of three replicates.

### 2.5. Antifungal Activity

#### 2.5.1. Agar Diffusion Method

Antifungal activity was evaluated against five strains isolated from the environment, food, or dairy products. They were chosen since they represent some of the most abundant fungal species causing significant food contamination [4]. The fungal strains were: *Aspergillus niger* (3071-13), *Paecilomyces* spp. (5332-9a), *Mucor racemosus* (LMA-722), *Penicillium crustosum* (27,159) and *Rhodotorula mucilaginosa* (27,173). The recovery method of the microorganisms from −80 °C, cell concentration count and the agar diffusion assay were described by Abou-Diab et al. [11]. As indicated by the Clinical and Laboratory Standards Institute (CLSI 2016, CLSI 2017), antifungal activity was measured as the diameter of the clear zone of growth inhibition and recorded as diameter of inhibition in millimeters. Natamycin (16.7 µg/mL) from Sigma-Aldrich (Oakville, ON, Canada) was used as positive control and sterile distilled water as negative control. All data are expressed as mean ± SD and are the mean of three replicates.

#### 2.5.2. Minimum Inhibitory Concentration (MIC)

The MIC assays were also applied to fungi and performed in sterile 96-well microplates. Each well contained 125 μL of peptone solution and was supplemented with 125 μL of hemoglobin hydrolysate solution at a concentration of 80 mg/mL serially diluted in half. Then, 50 μL of the mold and yeast strains (×10^5^ CFU/mL) was added to each well. The inhibition capacity of the peptides was compared to that of the well of a positive control consisting of peptone solution containing the strain and to that of the well of a negative control consisting of peptone solution. MIC was determined as the lowest concentration of peptides that caused no visible increase after 48 h of incubation at 25 °C. MIC values were done in triplicate. All data are expressed as mean ± SD and are the mean of three replicates. Minimal fungicidal concentrations (MFC) were determined by spotting 10μL from each well showing absence of growth onto Dichloran Rose–Bengal Chloramphenicol agar (DRBC, BD-Difco, Sparks, MD, USA) plates. After incubation at 25 °C for 48 h, the lowest dilution with no growth indicated the MFC. A compound was considered fungicidal when the MFC/MIC ratio was ≤4 and fungistatic when the MFC/MIC ratio was >4 [17].

### 2.6. Antioxidant Activity

#### 2.6.1. Antioxidant Assay Using the β-Carotene Bleaching Method

The ability of the different hydrolysates produced by EDBM and the control to prevent the bleaching of β-carotene was determined as described by Koleva et al. [18]. An emulsion of 4 mg β-carotene in 4 mL chloroform, mixed with 800 μL of Tween-40 and 100 μL of linoleic acid, was freshly prepared before each experiment. The chloroform was completely evaporated under vacuum (rotary evaporator, Heidolph, Schwabach, Germany) at 45 °C, then 400 mL of distilled water was added. An aliquot (5 mL) of the β-carotene–linoleic acid emulsion was transferred to tubes containing 500 μL of each sample at different concentrations. The tubes were placed in a water bath and incubated at 50 °C for 2 h. Thereafter, the absorbance of each sample was measured at 470 nm. Distilled water (500 μL) was used as a blank. Butylated hydroxytoluene (BHT, 0.5 mg/mL) was used as positive control. Relative antioxidant activity (*RAA%*) was calculated according to Equation (1):(1)$$RAA\left(\%\right)=\frac{A_{sample}}{A_{control}}\times 100$$
where *A_sample_* is the absorbance of samples (with the emulsion) and *A_control_* is the absorbance of BHT (with the emulsion). All data are expressed as mean ± SD and are the mean of three replicates.

#### 2.6.2. Antioxidant Assay Using DPPH Radical Scavenging Capacity

The DPPH radical-scavenging capacity of samples was determined as described by Bersuder et al. [19]. This antioxidant test is based on evaluating the reducing agents of the peptides in a colorimetric redox reaction. A mixture of each sample at different concentrations ((500 μL), 99% ethanol (375 μL) and DPPH solution (125 μL of 0.02% in ethanol)) were shaken and then incubated for 60 min in a dark room at 30 °C. Scavenging capacity was measured spectrophotometrically (UV Mini 1240, UV–Vis Spectrophotometer, SHIMDZU, Kyoto, Japan) by monitoring the decrease in absorbance at 517 nm. Lower absorbance of the reaction mixture indicated higher DPPH free radical-scavenging activity, since in its radical form DPPH has an absorption band at 517 nm which disappears upon reduction by an antiradical compound. DPPH radical-scavenging capacity was calculated as follows:(2)$$\mathrm{DPPH}\;\mathrm{radical}\;\mathrm{scavenging}\;\mathrm{activity}\;(\%)=\frac{A_{control}-A_{sample}}{A_{control}}\times 100$$
where *A_control_* represents the absorbance of all the reagents without the samples and *A_sample_* is the absorbance of the sample (with the DPPH solution). DPPH radical scavenging activity was expressed as half-maximal inhibition concentrations (IC_50_) and Trolox equivalent antioxidant capacity (TEAC) values. The IC_50_ was calculated graphically as described by Abou-Diab et al. [11]. Compounds are more anti-free-radical when the IC_50_ is low [20]. The TEAC value was defined as the concentration of standard Trolox that exhibited the same antioxidant capacity as a 1 mg/mL solution concentration of the antioxidant compound under investigation [21]. All data are expressed as mean ± SD and are the mean of three replicates.

#### 2.6.3. Antioxidant Assay Using ABTS Radical Scavenging Capacity

As described by Re et al. [22] and based on a colorimetric redox reaction, the reducing agents of the peptides were evaluated using the ABTS radical scavenging activity method. Between 12 and 16 h prior to use, a mixed solution of 7 mM ABTS and 4.95 mM potassium persulfate was prepared to obtain the cationic radical ABTS^+^. The solution was stored at room temperature and protected from light. Before use, an absorbance of less than 1 at 734 nm is required. Hence, a portion of this solution was diluted with ethanol at 30 °C. Then, 10 μL of each sample (hydrolysates were previously dissolved in ultrapure water at different concentrations) was mixed with 1 mL of the diluted ABTS^+^ solution. After 6 min of mixing, the absorbance of ABTS^+^ was measured at 734 nm, 30 °C using a Shimadzu UV-1650 PC Spectrophotometer (Shimadzu Corporation, Kyoto, Japan). Appropriate solvent blanks were run in each assay. The results are expressed as percentage of inhibition according to the following Equation (3):(3)$${\mathrm{ABTS}}^{+}\;\mathrm{radical}\;\mathrm{scavenging}\;\mathrm{activity}\;(\%)=(1-\frac{A_{sample}}{A_{control}})\times 100$$
where *A_control_* represents the absorbance of all the reagents without the samples and *A_sample_* is the absorbance of the sample (with the ABTS solution). ABTS^+^ radical scavenging activity was expressed as the half-maximal inhibition concentration (IC_50_) and TEAC values. All data are expressed as mean ± SD and are the mean of three replicates.

#### 2.6.4. Evaluation of Total Antioxidant Capacity

As described by Prieto et al. [23], the total antioxidant capacity—based on the reduction of molybdenum Mo present as molybdate MoO_4_^2−^ ions to molybdenum MoO^2+^ and forming a green phosphate/molybdenum (Mo_3_O_16_P_4_) complex—is a simple method to determine the antioxidant activity of the peptide hydrolysates at different concentrations (2.5, 5, 10 and 20 mg/mL). Briefly, an aliquot of 300 μL of each sample was mixed with 3 mL reagent (0.6 M sulfuric acid, 28 mM sodium phosphate, and 4 mM ammonium molybdate). The tubes were capped and incubated in a thermal block at 95 °C for 90 min. After cooling at room temperature, the absorbance of the aqueous solution of each sample was measured at 695 nm (Shimadzu UV-1650 PC Spectrophotometer, Shimadzu Corporation, Kyoto, Japan), against a blank. For calibration, a Trolox solution (0 to 1 mg/mL) was used, and the antioxidant activity was expressed in TEAC values using Equation (4) given by the calibration line:(4)$$A=a\times [C_{\mathrm{Trolox}}]+b$$
where A is the absorbance at 695 nm, and C is the equivalent antioxidant concentration (mg/mL). The concentration is determined according to the equation of the standard range curve of a reference antioxidant such as Trolox, with a and b, respectively, the origin and the slope of the Trolox calibration line. All data are expressed as mean ± SD and are the mean of three replicates.

### 2.7. Statistical Analyses

All analyses were performed in triplicate, and three independent repetitions were done for each condition. Data were subjected to one-way or two-way analyses of variance (ANOVA). Tukey tests were also performed on data using SigmaPlot software (version 14.0) to determine which treatments were statistically different from the others at a probability level *p* of 0.05.

## 3. Results and Discussion

### 3.1. Antibacterial Activity

#### 3.1.1. Agar Diffusion Method

Antibacterial activity investigated on CH-RH, CH-DH, EDBM-RH and EDBM-DH at 20 mg/mL are presented in Table 1. All hemoglobin hydrolysates derived from EDBM and conventional hydrolysis displayed antibacterial activity against the six tested bacterial species, with a clear growth inhibition zone. There is no significant difference in antibacterial activity between CH-RH, CH-DH and EDBM-DH (20 mg/mL) against *Staphylococcus aureus* and *Kocuria rhizophila*, since the zones of inhibition of the three conditions are >1.5 cm (+++). However, EDBM-RH presented a smaller and statistically lower (*p* < 0.05) inhibition zone than the other hydrolysates. Moreover, no significant difference between hydrolysates was observed against the other strains. *Staphylococcus aureus*, *Kocuria rhizophila* and *Listeria monocytogenes* presented the highest sensitivity to the hydrolysates.

The inhibition zones induced by CH-RH, CH-DH, EDBM-RH and EDBM-DH at 10 and 20 mg/mL against *Staphylococcus aureus* and *Kocuria rhizophila* are presented in Figure 3. Hydrolysates at 20 mg/mL induced larger inhibition zones than hydrolysates at 10 mg/mL.

#### 3.1.2. Minimum Inhibitory Concentration (MIC)

MIC values of the hydrolysates presented in Table 2 are expressed in mg/mL due to the presence of a large peptide population, as demonstrated by Abou-Diab et al. [13]. The results of MIC confirmed the findings previously demonstrated by agar diffusion. Lower MIC values of hydrolysates were registered against *Staphylococcus aureus, Kocuria rhizophila* and *Listeria monocytogenes* (between 0.31 to 2.5 mg/mL). However, higher MIC values of hydrolysates were registered against *Micrococcus luteus*, *Escherichia coli* and *Salmonella newport* (between 5 to 10 mg/mL).

The antibacterial effects of CH-RH, CH-DH and EDBM-DH were statistically similar (*p* > 0.05) and corresponded to MIC values of 0.31 mg/mL against *Staphylococcus aureus* and *Kocuria rhizophila*, while EDBM-RH was statistically different (*p* < 0.05) and corresponded to MIC values of 1.25 and 0.62 mg/mL against *Staphylococcus aureus* and *Kocuria rhizophila*, respectively. CH-RH displayed important and statistically lower (*p* < 0.05) antibacterial activity than the others against *Listeria monocytogenes* and *Salmonella newport* with MIC values of 1.25 and 5 mg/mL, respectively. The MIC value of 5 mg/mL against *Micrococcus luteus* obtained for CH-RH and CH-DH was statistically different (*p* < 0.05) from the MIC value of 10 mg/mL for EDBM-RH and EDBM-DH. Identical antibacterial activity was also recorded for all hydrolysates against *Escherichia coli* with a MIC value of 10 mg/mL (Table 2). Our findings are in line with previous findings reported by Abou-Diab et al. [11] and Zouari et al. [24], who showed the inhibitory effect of bovine or porcine hemoglobin hydrolysates against *Staphylococcus aureus, Kocuria rhizophila*, *Listeria monocytogenes*, *Micrococcus luteus*, *Escherichia coli* and *Salmonella newport*.

### 3.2. Antifungal Activity

#### 3.2.1. Agar Diffusion Method

Antifungal activity was investigated on CH-RH, CH-DH, EDBM-RH and EDBM-DH at two different concentrations (10 and 20 mg/mL). The hydrolysates were tested against *Paecilomyces* spp. (5332-9a), *Aspergillus niger* (3071-13), *Mucor racemosus* (LMA-722), *Penicillium crustosum* (27,159) and *Rhodotorula mucilaginosa* (27,173). A large growth inhibition zone was observed against *Paecilomyces* spp. and *Rhodotorula mucilaginosa* (Figure 4). CH-RH, CH-DH and EDBM-DH at a concentration of 20 mg/mL were strongly active against *Paecilomyces* spp., inducing inhibition zones of 18 mm, 18.5 mm and 18 mm, respectively, while CH-RH, CH-DH, and EDBM-DH, at a concentration of 10 mg/mL, induced inhibition zones of 14 mm, 14.5 mm and 13.5 mm, respectively. Statistical analysis demonstrated that there was no significant difference (*p* > 0.05) between hydrolysates having the same concentration. However, EDBM-RH presented a smaller and statically lower (*p* < 0.05) inhibition zone than the other hydrolysates (14.5 mm at 20 mg/mL and 11 mm at 10 mg/mL). Statistical analysis demonstrated that there was no significant difference (*p* > 0.05) in the inhibition zone formed by CH-RH, CH-DH and EDBM-DH against *Rhodotorula mucilaginosa*. At 20 mg/mL, CH-RH presented an inhibition zone (18.5 mm) roughly larger than CH-DH (18 mm) and even roughly larger than EDBM-DH (17 mm). At 10 mg/mL, CH-RH presented an inhibition zone (14 mm) similar to that of CH-DH (14 mm) and roughly larger that EDBM-DH (13 mm). While a statistical difference (*p* < 0.05) in the inhibition zone formed by EDBM-RH was noticed (13.5 mm at 20 mg/mL and 10 mm at 10 mg/mL). All hydrolysates at 20 mg/mL induced an inhibition zone against *Rhodotorula mucilaginosa* statistically higher (*p* < 0.05) than the positive control (11 mm). At a concentration of 20 mg/mL, CH-RH presented clear activity against *Mucor racemosus* (inhibition zone of 15 mm) statistically similar (*p* > 0.05) to CH-DH (14 mm) and EDBM-DH (13 mm) but statistically higher (*p* < 0.05) than that of EDBM-RH (11 mm). At 10 mg/mL, all hydrolysates were active against *Mucor racemosus*, inducing roughly similar inhibition zones of 8, 8, 6 and 7 mm for CH-RH, CH-DH, EDBM-RH and EDBM-DH, respectively. CH-RH, CH-DH, EDBM-RH and EDBM-DH at a concentration of 20 mg/mL were active against *Penicillium crustosum*, producing inhibition zones of 10, 11, 8 and 9 mm, respectively, and statistically higher (*p* < 0.05) than the inhibition zone at 10 mg/mL. Moreover, CH-RH, CH-DH, EDBM-RH and EDBM-DH at a concentration of 20 mg/mL were active against *Aspergillus niger*, producing inhibition zones of 10, 11, 8 and 9 mm, respectively, and statistically higher (*p* < 0.05) than the inhibition zone at 10 mg/mL in comparison with the negative control, which had no inhibition zone. The smaller inhibition zone noticed for EDBM-RH may be due to the lower peptide population recovered at the end of the process due to the precipitation of heme or peptide-heme associations in the diluate compartment as explained by Abou-Diab et al. [13].

#### 3.2.2. MIC and MFC of Bovine Hemoglobin Hydrolysates

MICs of CH-RH, CH-DH, EDBM-RH and EDBM-DH against *Paecilomyces* spp. (5332-9a), *Aspergillus niger* (3071-13), *Mucor racemosus* (LMA-722), *Penicillium crustosum* (27,159) and *Rhodotorula mucilaginosa* (27,173) were determined by broth microdilution followed by visual reading (Table 3). All samples showed activity against the different fungal strains tested. The results of MIC confirmed the findings previously obtained with by agar diffusion. The MIC values recorded for hydrolysates against *Paecilomyces* spp. and *Rhodotorula mucilaginosa* (between 0.44 to 3.5 mg/mL) were considerably lower than the MIC values registered against *Mucor racemosus*, *Penicillium crustosum* and *Aspergillus niger* (between 3.57 to 7.15 mg/mL). The antifungal effect of CH-RH, CH-DH and EDBM-DH exhibited a broad spectrum of activity, inhibiting *Paecilomyces* spp. and *Mucor racemosus* with statistically similar (*p* > 0.05) MIC values of 0.44 and 3.57 mg/mL, respectively. However, EDBM-RH exhibited a higher and statistically different (*p* < 0.05) MIC value to inhibit *Paecilomyces* spp. (0.89 mg/mL) and *Mucor racemosus* (7.15 mg/mL). EDBM-RH also showed a higher and statistically different (*p* < 0.05) MIC value of 3.57 mg/mL against *Rhodotorula mucilaginosa.* The antifungal activity of CH-RH, CH-DH, EDBM-RH and EDBM-DH, against *Penicillium crustosum* and *Aspergillus niger* showed a statistically simi- lar (*p* > 0.05) MIC value of 7.15 mg/mL.

MFC of bovine hemoglobin hydrolysates ranged between 0.44 and 7.15 mg/mL for all fungal strains except *Penicillium crustosum* and *Aspergillus niger,* which did not display any MFC value. As described by Pfaller et al. [18] by extrapolation from the conventional definition used for bacterial testing, CH-RH, CH-DH, EDBM-RH and EDBM-DH acted with a fungicidal mechanism towards *Paecilomyces* spp., *Rhodotorula mucilaginosa* and *Mucor racemosus* since the MFC/MIC ratio was ≤4. Moreover, the hydrolysates only inhibited the growth of *Penicillium crustosum* and *Aspergillus niger* after 48 h since the MFC/MIC ratio was not determined.

These results demonstrated that bovine hemoglobin hydrolysates, whatever their mode of production, had antibacterial and antifungal activities. Moreover, the demineralization-discoloration did not impact the antimicrobial capacity. However, a lower antibacterial and antifungal activity was noticed for EDBM-RH. This can be explained by the fact that this hydrolysate exhibited fewer antibacterial peptides, as demonstrated by Abou-Diab et al. [13], who identified a lower peptide population in EDBM-RH compared to other hydrolysates due to the precipitation of some peptides during demineralization in the diluate compartment: α1-29, α1-28 and α1-27, which are known for their antibacterial activity, appeared in all conditions and were not charged at pH 7; α33-46, α34-46, α37-46, α99-105, α99-106, β126-145 and β140-145, which are known for their antibacterial activity, appeared in all conditions and were charged +1 at pH 7; α137-141 appeared in all conditions and was charged +2 at pH 7; α34-66, α33-45, α133-141 and α34-65, which are known for their antibacterial activity and whose overall charge at pH 7 was +2, did not appear in EDBM-RH but did appear in CH-RH, CH-DH and EDBM-DH. Therefore, the antibacterial and antifungal capacities of EDBM-RH were lower since at the same pH (adjusted to 7) of the final hydrolysates, the overall charge of antibacterial peptides identified in EDBM-RH had a value of +7, lower than the overall charge of antibacterial peptides identified in CH-RH, CH-DH and EDBM-DH, with values of +15, +14 and +14, respectively.

The antibacterial peptides are the most characterized because the focus on discovery of antimicrobial peptides has long been restricted to those with antibacterial activity. However, the mechanism of action of antibacterial and antifungal activities is roughly the same, as reported by Yeaman et al. [25], Van Der Weerden et al. [26], Bahar et al. [27] and Kumar et al. [28]. The charge, conformation, hydrophobicity and amphiphilic character of the peptide sequence are all elements influencing the antimicrobial potential [25,26,27,28]. The charge of the antimicrobial peptide is an essential part of its activity. In general, it is described that cationic domains are essential for the establishment of interactions between the biological, negatively charged phospholipid membrane and the peptide through electrostatic interactions. The higher the charge of the peptide, the greater the antimicrobial potential of the peptide [25,26,27,28].

### 3.3. Antioxidant Activity

#### 3.3.1. Antioxidant Assay Using β-Carotene Bleaching

This method is based on the loss of the yellow color of β-carotene due to its reaction with radicals formed by linoleic acid oxidation in an emulsion. β-carotene bleaching, measured by the decrease in the initial absorbance at 470 nm, is slowed in the presence of antioxidants. The method is widely used in the antioxidant activity evaluation of different types of compounds, as described by Koleva et al. [18]. In this study, the inhibition activity of lipid peroxidation was determined by assessing the ability of CH-RH, CH-DH, EDBM-RH and EDBM-DH to inhibit the oxidation of linoleic acid in this emulsified model.

The antioxidant activities of CH-RH, CH-DH, EDBM-RH and EDBM-DH at different concentrations using the β-carotene bleaching assay are presented in Figure 5. The results were compared to the standard antioxidant, BHT (0.5 mg/mL), and synthetic bioactive peptide, neokyotorphin (NKT, 0.5 mg/mL). The oxidation of β-carotene was inhibited by CH-RH, CH-DH, EDBM-RH and EDBM-DH. However, different relative antioxidant activities (%) were noticed depending on concentration. A concentration of 20 mg/mL showed significantly (*p* < 0.05) higher antioxidant activity than the other concentrations. At concentrations of 2.5 and 20 mg/mL there was no significant difference (*p* > 0.05) in antioxidant activity between CH-RH, CH-DH, EDBM-RH and EDBM-DH, while at concentrations of 5 and 10 mg/mL there was a significant difference (*p* < 0.05) in antioxidant activity, but the values were very close. The antioxidant activity of CH-RH, CH-DH, EDBM-RH and EDBM-DH at a concentration of 20 mg/mL (90.92 ± 1.07; 89.93 ± 0.13; 89.73 ± 1.55; 91.49 ± 0.42%, respectively) were significantly lower (*p* < 0.05) but very close to that of BHT (99.00 ± 0.31%) and NKT (94.41 ± 1,03%). Bleaching of β-carotene was slowed in the presence of bovine hemoglobin hydrolysates. Therefore, CH-RH, CH-DH, EDBM-RH and EDBM-DH could be considered as antioxidants protecting food from lipid peroxidation.

#### 3.3.2. DPPH Radical Scavenging Capacity

DPPH radicals have been widely used to evaluate antioxidant capacity based on the ability of molecules to transfer one electron to reduce an oxidant. The DPPH radical sca- venging activity method was performed on CH-RH, CH-DH, EDBM-RH and EDBM-DH at different concentrations. DPPH radical scavenging activities of CH-RH, CH-DH, EDBM-RH and EDBM-DH were expressed as the half-maximal inhibition concentrations (IC_50_) and Trolox equivalent antioxidant capacity (TEAC). The comparative IC_50_ and TEAC values of the tested phenols were shown in Table 4.

The DPPH radical scavenging activities of CH-RH, CH-DH, EDBM-RH and EDBM-DH were different (*p* < 0.05) for IC_50_ and lower than Trolox. However, NKT showed an IC_50_ close to that of Trolox but statistically lower (*p* < 0.05). The TEACs of EDBM-RH and EDBM-DH were statistically similar (*p* > 0.05) but higher (*p* < 0.05) than CH-RH and CH-DH. In comparison with Trolox, the DPPH radical scavenging activity of all the hydrolysates expressed in TEAC showed lower radical scavenging activity (TEAC value lower than 1). As expected, the TEAC of NKT showed higher radical scavenging activity than the hydrolysates, with a value closer to 1. These findings are consistent with recent work reported by Abou-Diab et al. [11] where bovine hemoglobin hydrolysates showed important DPPH scavenging capacity but lower capacity compared to Trolox due to the competition effect of bioactive peptides present in large numbers in the hydrolysates.

#### 3.3.3. Antioxidant Property Products by ABTS Assay

The ABTS assay is a widely applied method for measuring the radical scavenging ability of antioxidants, especially those present in foods. The antioxidant capacities of different concentrations CH-RH, CH-DH, EDBM-RH and EDBM-DH were measured for their quenching capacity toward the ABTS radical cation. The results presented in Figure 6 clearly indicate that hydrolysates produced using EDBM exhibited strong ABTS radical scavenging activity. The antioxidant activity of bovine hemoglobin hydrolysates increased with increasing peptide concentrations. A concentration of 20 mg/mL for CH-RH, CH-DH, EDBM-RH and EDBM-DH showed significantly higher (*p* < 0.05) scavenging capacity than the other concentrations. At a concentration of 10 and 20 mg/mL, there was no significant difference (*p* > 0.05) in ABTS scavenging capacity between CH-RH, CH-DH, EDBM-RH and EDBM-DH, while at a concentration of 2.5 and 5 mg/mL there was a significant difference (*p* < 0.05) in ABTS scavenging capacity between CH-RH/DH and EDBM-RH/DH. Our findings are in line with previous work by Abou-Diab et al. [11], who reported that ABTS scavenging activity increased with increasing peptide hydrolysate concentrations (2.5–20 mg/mL).

Trolox was used as a reference molecule. The comparative IC_50_ and the relative values expressed as TEAC are reported in Table 5. To inhibit 50% of ABTS radicals, concentrations of hemoglobin hydrolysates between 3.43 and 4 mg/mL were needed. Statistical analysis showed that there was a significant difference (*p* < 0.05) for IC_50_ for bovine hemoglobin hydrolysates between all conditions even if the values were very similar. To inhibit 50% of ABTS radicals, a concentration of 0.56 mg/mL of NKT was needed; this value is very close to Trolox but is statistically different (*p* < 0.05). Regarding TEAC, the ABTS radical scavenging activities of CH-RH and CH-DH were similar but statistically lower (*p* < 0.05) than the ABTS radical scavenging activities of EDBM-RH and EDBM-DH. However, TEAC values of all hydrolysates were lower than the TEAC value of Trolox. TEAC of NKT (0.89 ± 0.03) was close to Trolox.

#### 3.3.4. Evaluation of Total Antioxidant Capacity

The capacity of CH-RH, CH-DH, EDBM-RH and EDBM-DH as a reducing agent to produce Mo(V) from Mo(VI) and to form a green-colored complex based on an oxidation–reduction reaction was carried out at different concentrations using the total antioxidant capacity method and expressed as TEAC (Figure 7).

The antioxidant capacity increased with increasing concentration. The TEACs of CH-RH, CH-DH, EDBM-RH and EDBM-DH increased from 0.15 ± 0.00; 0.12 ± 0.01; 0.15 ± 0.01; 0.15 ± 0.00 mg/mL, respectively, at a concentration of 2.5 mg/mL, to 0.31 ± 0.00; 0.31 ± 0.00; 0.33 ± 0.01; 0.33 ± 0.03 mg/mL at a concentration of 20 mg/mL, respectively. As expected, a concentration of 20 mg/mL showed significantly higher antioxidant activity than the other concentrations with the highest TEAC value. For the same concentration of 5, 10 and 20 mg/mL there was no significant difference (*p* > 0.05) in TEAC between CH-RH, CH-DH, EDBM-RH and EDBM-DH. Meanwhile, for the same concentration of 2.5 mg/mL, there was a significant difference (*p* < 0.05) in TEAC between CH-RH and CH-DH. Based on this test, bovine hemoglobin hydrolysates could be considered an effective reducing agent and thus a primary antioxidant even at high temperature (95 °C).

After performing four antioxidant assays based on different mechanisms, these results demonstrate that bovine hemoglobin hydrolysates, whatever their mode of production, had antioxidant capacities. The important antioxidant activity of CH-RH, CH-DH, EDBM-RH and EDBM-DH was based on the ability of peptides to reduce free radicals by hydrogen donation in a competitive reaction and/or their ability to transfer one electron to reduce an oxidant. It is important to mention that demineralization and discoloration did not impact the antioxidant capacity; whatever the method of hydrolysate production, the antioxidant activity was roughly the same. This is due to the resemblance in the peptide population between the conditions, as described by Abou-Diab et al. [13]. Moosmann and Behl [29] and Stadtman and Levine [30] reported that two categories of amino acids (AA) are well known for their antioxidant activities: phenolic AAs such as tyrosine, tryptophan or phenylalanine, and reducing AAs (sulfur-containing) such as methionine or cysteine. In our study based on roughly the same peptide population, it is possible to have the same phenolic and reducing amino acid content and, therefore, the same antioxidant activity. The bioactive peptide sequences containing the phenolic and/or the reducing AA as: TSKYR, LSFPTTKTYFPHF and VLSAADKGNVKAAWGKVGGHAAEYGAE were identified in CH-RH, CH-DH, EDBM-RH and EDBM-DH as reported by Abou-Diab et al. [13]. Hydrolysates derived from EDBM are more interesting at the industrial scale since they are pure peptides with 85% less salt. Although the chemical antioxidant standards exhibited the highest antioxidant ability, natural antioxidants are of growing interest. Indeed, the incorporation of a natural protein hydrolysate with low salt content to foods could confer desirable nutritional and functional properties [31].

## 4. Conclusions

In this study, we tested, for the first time, the biological activity of bovine hemoglobin demineralized and discolored–demineralized hydrolysates produced by EDBM. The results indicate that CH-RH, CH-DH and EDBM-DH showed excellent bacterial growth inhibition against *Listeria monocytogenes*, *Staphylococcus aureus*, *Micrococcus luteus*, *Escherichia coli*, *Kocuria rhizophila* and *Salmonella newport*. Furthermore, they showed excellent fungal growth inhibition against *Paecilomyces* spp., *Aspergillus niger*, *Mucor racemosus*, *Penicillium crustosum* and *Rhodotorula mucilaginosa*. However, EDBM-RH showed lower antibacterial and antifungal activities compared to other hydrolysates due to loss of antimicrobial peptides after precipitation in the diluate compartment. Moreover, these results confirmed that CH-RH, CH-DH, EDBM-RH and EDBM-DH, whatever their mode of production, had antioxidant capacities. The demineralization and discoloration did not affect the antioxidant potential. The hydrolysates can act as inhibitors of lipid peroxidation, as direct scavengers of free radicals and as agents to chelate the transition-metal ions that catalyze the generation of radical species.

The production of bovine hemoglobin discolored–demineralized hydrolysates by the new optimized multiple-step EDBM process described by Abou-Diab et al. [13] fits perfectly within the concept of circular economy and could have great importance at the industrial level. Since bovine hemoglobin is derived from slaughterhouse blood, a natural bioresource produced in very large quantities that represents the major protein part of the blood, it was valorized by this new process as a representative product before generalization to whole blood. Indeed, bovine hemoglobin from blood, a slaughterhouse waste product, once hydrolyzed, allowed the bioproduction of pure, active, discolored peptides with a low concentration of mineral salts that could be used as bio-preservative alternatives to chemical additives. Therefore, they could be promising candidates to improve food safety and avoid food spoilage. However, further studies are required to investigate the in vivo effects of these hydrolysates on meat during food storage.

## Figures and Tables

**Figure 1 membranes-12-00512-f001:**
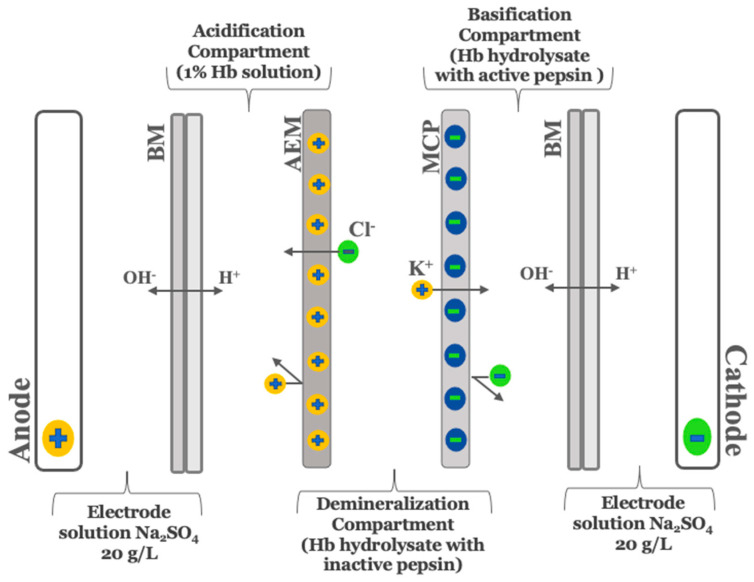
Schematic representation of three-compartment bipolar membrane electrodialysis cell configuration for production of demineralized and discolored–demineralized hemoglobin hydrolysates running simultaneously used by Abou-Diab et al. [13]. BM, bipolar membrane; MCP, monovalent cation perm-selective membrane; AEM, anion exchange membrane.

**Figure 2 membranes-12-00512-f002:**
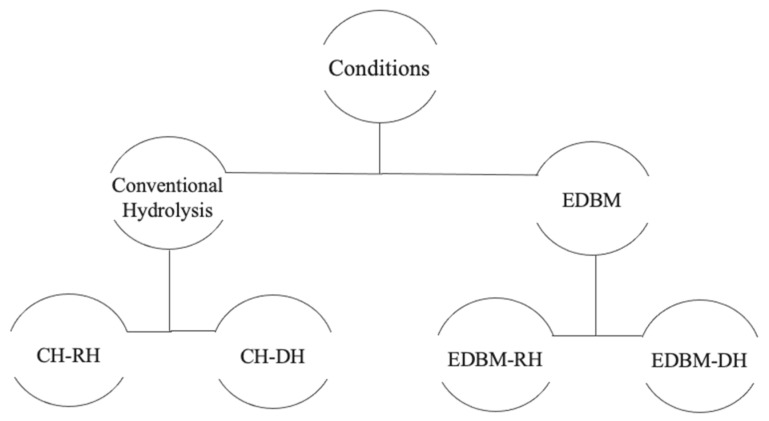
Diagram summarizing the hydrolysates recovered in each condition.

**Figure 3 membranes-12-00512-f003:**
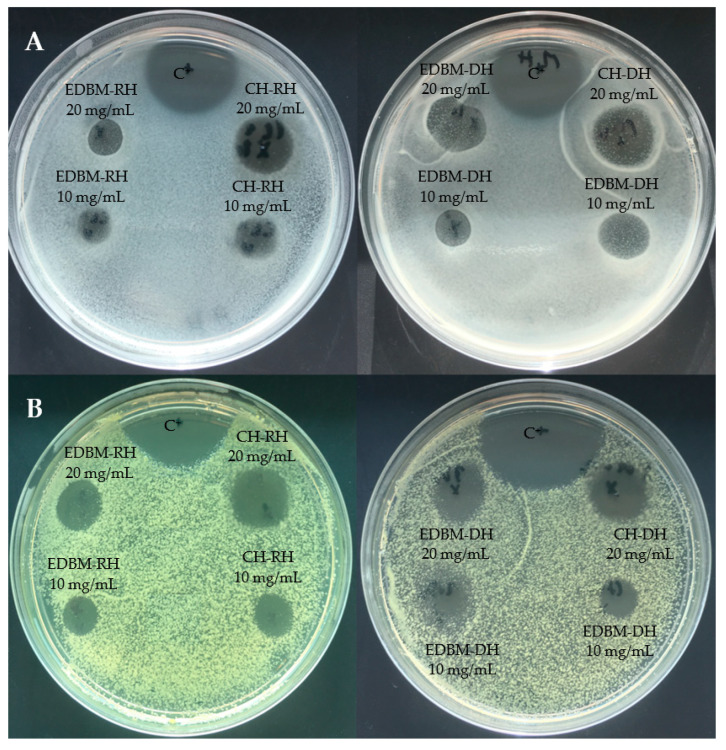
Inhibition zones induced by 2 different concentrations (10 and 20 mg/mL) of raw and discolored bovine hemoglobin hydrolysates produced by two different hydrolysis processes on Muller-Hinton Agar overlaid with a suspension (10^6^ strain per plate) of *Staphylococcus aureus* (**A**) and *Kocuria rhizophila* (**B**) and compared to a positive control (C^+^: ampicillin).

**Figure 4 membranes-12-00512-f004:**
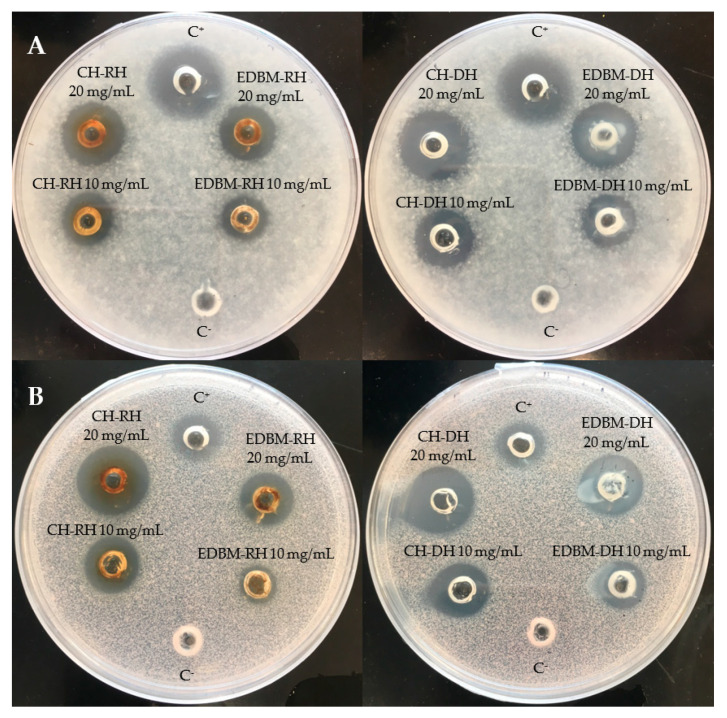
Inhibition zones induced by 2 different concentrations (10 and 20 mg/mL) of raw and discolored bovine hemoglobin hydrolysates produced by two different hydrolysis processes on potato dextrose agar (PDA) overlaid with a suspension (10^4^ spores per plate) of *Paecilomyces* spp. (**A**) and *Rhodotorula mucilaginosa* (**B**) and compared to a positive (C^+^: natamycin) and negative control (C^−^: distilled water).

**Figure 5 membranes-12-00512-f005:**
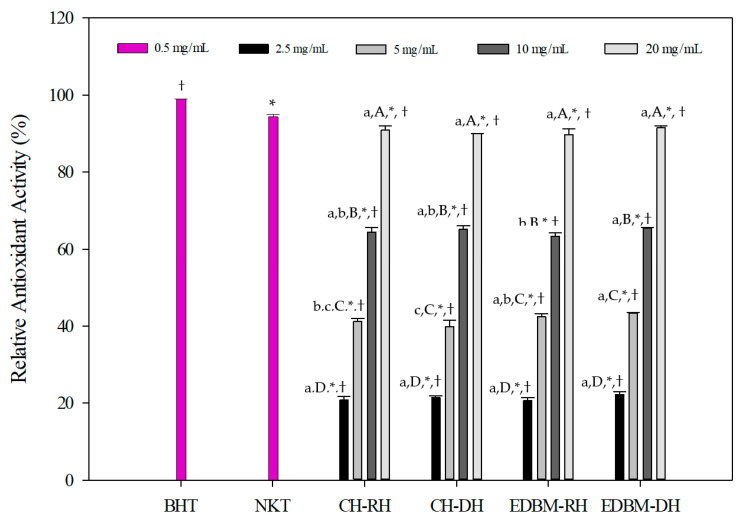
β-carotene bleaching inhibition activity of bovine hemoglobin hydrolysates at different concentrations in CH-RH, CH-DH, EDBM-RH and EDBM-DH. Values with (*) symbol are significantly different from BHT; values with (†) symbol are significantly different from neokyotorphin; values with different lowercase letters (a–c) within the same concentration are significantly different; values with different capital letters (A–D) within the same condition are significantly different (*p* < 0.05, Tukey test).

**Figure 6 membranes-12-00512-f006:**
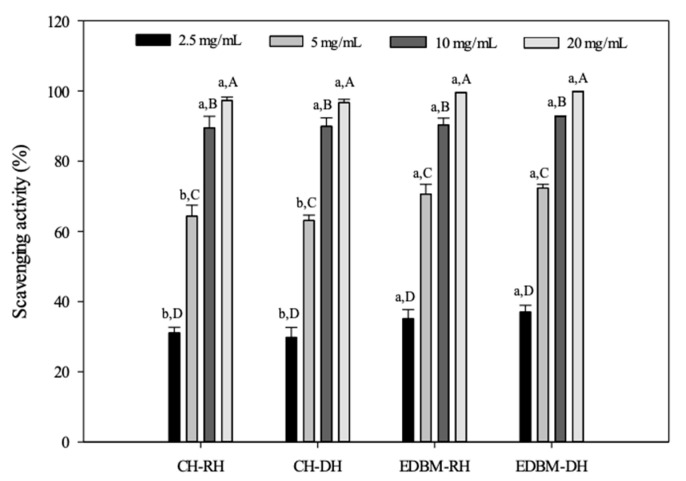
ABTS scavenging activities of bovine hemoglobin hydrolysates in CH-RH, CH-DH, EDBM-RH and EDBM-DH at different concentrations. Values with different lowercase letters (a,b) within the same concentration are significantly different; values with different capital letters (A–D) within the same condition are significantly different (*p* < 0.05, Tukey test).

**Figure 7 membranes-12-00512-f007:**
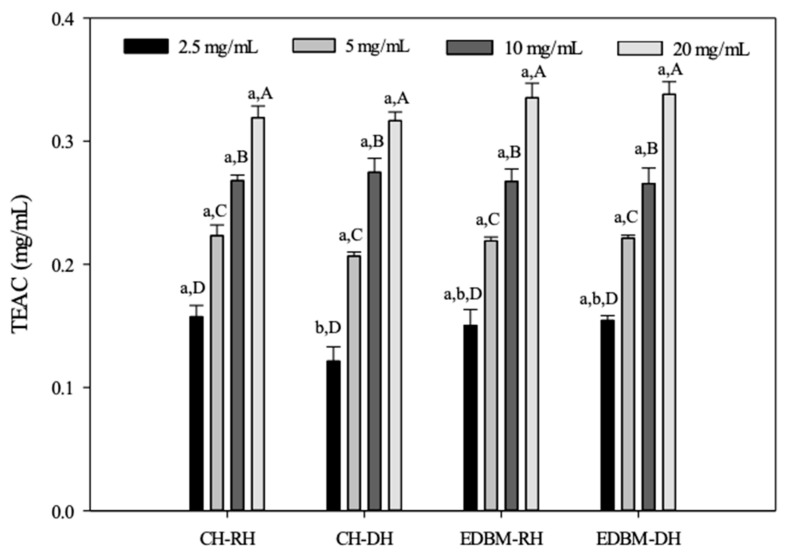
Total antioxidant capacity of bovine hemoglobin hydrolysates in CH-RH, CH-DH, EDBM-RH and EDBM-DH at different concentrations. Values with different lowercase letters (a,b) within the same concentration are significantly different; values with different capital letters (A–D) within the same condition are significantly different (*p* < 0.05, Tukey test).

**Table 1 membranes-12-00512-t001:** Antibacterial activity of bovine hemoglobin hydrolysates.

Bacteria Strains	Bovine Hemoglobin Hydrolysates			
	CH-RH	CH-DH	EDBM-RH	EDBM-DH
*Staphylococcus aureus*	+++	+++	++	+++
*Listeria monocytogenes*	++	++	++	++
*Micrococcus luteus*	+	+	+	+
*Kocuria rhizophila*	+++	+++	++	+++
*Escherichia coli*	+	+	+	+
*Salmonella newport*	+	+	+	+

Inhibition zones: +++: >1.5 cm; ++: 0.5–1.5 cm; +: <0.5 cm.

**Table 2 membranes-12-00512-t002:** MIC values of the bovine hemoglobin hydrolysates.

Bacteria Strains	Bovine Hemoglobin Hydrolysates			
	MIC			
	CH-RH	CH-DH	EDBM-RH	EDBM-DH
	mg/mL	mg/mL	mg/mL	mg/mL
*Staphylococcus aureus*	0.31 ± 0.0 ^b^	0.31 ± 0.0 ^b^	1.25 ± 0.0 ^a^	0.31 ± 0.0 ^b^
*Listeria monocytogenes*	1.25 ± 0.0 ^b^	2.5 ± 0.0 ^a^	2.5 ± 0.0 ^a^	2.5 ± 0.0 ^a^
*Micrococcus luteus*	5 ± 0.0 ^b^	5 ± 0.0 ^b^	10 ± 0.0 ^a^	10 ± 0.0 ^a^
*Kocuria rhizophila*	0.31 ± 0.0 ^b^	0.31 ± 0.0 ^b^	0.62 ± 0.0 ^a^	0.31 ± 0.0 ^b^
*Escherichia coli*	10 ± 0.0 ^a^	10 ± 0.0 ^a^	10 ± 0.0 ^a^	10 ± 0.0 ^a^
*Salmonella newport*	5 ± 0.0 ^b^	10 ± 0.0 ^a^	10 ± 0.0 ^a^	10 ± 0.0 ^a^

The minimum inhibitory concentration (MIC) of the peptide hydrolysates was determined in a microtiter plate assay system after 24 h of incubation at 37 °C. ^a,b^: Population means for each bacteria strain within each row with different letters are significantly different, similar letter means no significant difference *p* < 0.05 (Tukey test).

**Table 3 membranes-12-00512-t003:** In vitro MIC and MFC ranges of CH-RH, CH-DH, EDBM-RH and EDBM-DH against filamentous molds and yeast as determined by broth microdilution followed by visual examination.

Bovine Hemoglobin Hydrolysates (mg/mL)												
	CH-RH			CH-DH			EDBM-RH			EDBM-DH		
Fungal Strain	MIC	MFC	MFC/MIC	MIC	MFC	MFC/MIC	MIC	MFC	MFC/MIC	MIC	MFC	MFC/MIC
*Paecilomyces* spp.	0.44 ± 0.0 ^b^	0.44 ± 0.0 ^C^	1	0.44 ± 0.0 ^b^	0.44 ± 0.0 ^C^	1	0.89 ± 0.0 ^a^	0.89 ± 0.0 ^B^	1	0.44 ± 0.0 ^b^	1.78 ± 0.0 ^A^	4
*Aspergillus niger*	7.15 ± 0.0 ^a^	0.0 ± 0.0 ^A^	ND	7.15 ± 0.0 ^a^	0.0 ± 0.0 ^A^	ND	7.15 ± 0.0 ^a^	0.0 ± 0.0 ^A^	ND	7.15 ± 0.0 ^a^	0.0 ± 0.0 ^A^	ND
*Rhodotorula mucilaginosa*	0.89 ± 0.0 ^c^	1.78 ± 0.0 ^C^	2	1.78 ± 0.0 ^b^	7.15 ± 0.0 ^B^	4	3.57 ± 0.0 ^a^	14.3 ± 0.0 ^A^	4	1.78 ± 0.0 ^b^	7.15 ± 0.0 ^B^	4
*Mucor racemosus*	3.57 ± 0.0 ^b^	7.15 ± 0.0 ^A^	2	3.57 ± 0.0 ^b^	3.57 ± 0.0 ^B^	1	7.15 ± 0.0 ^a^	7.15 ± 0.0 ^A^	1	3.57 ± 0.0 ^b^	7.15 ± 0.0 ^A^	2
*Penicillium crustosum*	7.15 ± 0.0 ^a^	0.0 ± 0.0 ^A^	ND	7.15 ± 0.0 ^a^	0.0 ± 0.0 ^A^	ND	7.15 ± 0.0 ^a^	0.0 ± 0.0 ^A^	ND	7.15 ± 0.0 ^a^	0.0 ± 0.0 ^A^	ND

ND, not determined. For each strain, different lowercase letters indicate significant difference between conditions for MIC; different capital letters indicate significant difference between conditions for MFC; similar letters mean no significant difference, *p* < 0.05 (Tukey test).

**Table 4 membranes-12-00512-t004:** IC_50_ and Trolox equivalent antioxidant capacity (TEAC) coefficients of bovine hemoglobin hydrolysates for the DPPH method.

DPPH	CH-RH	CH-DH	EDBM-RH	EDBM-DH	NKT	Trolox
IC_50_ (mg/mL)	2.52 ± 0.01 ^b^	2.77 ± 0.03 ^a^	2.39 ± 0.09 ^c^	2.29 ± 0.04 ^d^	0.58 ± 0.02 ^e^	0.36 ± 0.01 ^f^
TEAC	0.14 ± 0.002 ^d^	0.13 ± 0.005 ^e^	0.15 ± 0.001 ^c^	0.15 ± 0.003 ^c^	0.64 ± 0.01 ^b^	1 ^a^

^a–f^: Mean values within each row with different letters are significantly different; similar letters mean no significant difference at a probability level of 0.05 (Tukey test).

**Table 5 membranes-12-00512-t005:** IC_50_ and TEAC coefficients of bovine hemoglobin hydrolysates for ABTS method.

ABTS	CH-RH	CH-DH	EDBM-RH	EDBM-DH	NKT	Trolox
IC_50_ (mg/mL)	3.91 ± 0.02 ^b^	4.00 ± 0.02 ^a^	3.54 ± 0.01 ^c^	3.43 ± 0.02 ^d^	0.56 ± 0.02 ^e^	0.50 ± 0.02 ^f^
TEAC	0.12 ± 0.004 ^d^	0.12 ± 0.005 ^d^	0.14 ± 0.005 ^c^	0.14 ± 0.005 ^c^	0.89 ± 0.03 ^b^	1 ^a^

^a–f^: Mean values within each row with different letters are significantly different, similar letters mean no significant difference *p* < 0.05 (ANOVA, Tukey test).

## Data Availability

The data presented in this study are available on request from the corresponding authors.

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
