# Peer review of "Production of Demineralized Antibacterial, Antifungal and Antioxidant Peptides from Bovine Hemoglobin Using an Optimized Multiple-Step System: Electrodialysis with Bipolar Membrane"

_membranes, 2022, doi:10.3390/membranes12050512_

Round 1

Reviewer 2 Report

The manuscript entitled "Production of demineralized Antibacterial, Antifungal and Antioxidant peptides from bovine hemoglobin using an optimized « multiple-steps » system: Electrodialysis with Bipolar Membrane"  has a good general idea but please consider the following comments:

1- The abstract needs more precise data.

2- Line 128 add a reference

3- Lines 338, 351, 377, 436, and 506 please add the numbers of tables and figures

Reviewer 3 Report

I suggest that the authors single out works on EDBM in a separate section so that the method for obtaining peptides and their difference from peptides obtained by other methods is clear. 

Figures 3 and 4 have unreadable inscriptions.

There are numerous errors in the numbering of the bibliography in the text. (Line 170, 351, 361, 377, 394, 506).

Round 2

Reviewer 1 Report

Peer review comments have been carefully addressed. 

Author Response

Thank you for taking the time to read our manuscript and thanks a lot for helping us improving it.  

Reviewer 3 Report

The authors have substantially revised the article. However, small errors in references remain (line 341, 355, 365, 398).
In my humble opinion, the article is more devoted to issues related to biotechnology than to membranes. But if the first part of the study is also published in the Membranes, it would be logical to publish the second part of the study here.

Author Response

The authors have substantially revised the article. However, small errors in references remain (line 341, 355, 365, 398).

Thank you for taking the time to read our manuscript and thanks a lot for helping us improving it. The modifications were done in the manuscript.

In my humble opinion, the article is more devoted to issues related to biotechnology than to membranes. But if the first part of the study is also published in the Membranes, it would be logical to publish the second part of the study here.

Thank you for your comment. Our first work concerning the feasibility study of electrodialysis with bipolar membranes has been published in membranes as well as the 2nd part concerning the biological activities. This present study is a continuation of our work using an optimized electrodialysis system with bipolar membranes based on previous studies. Thank you for your opinion and interest in our article but we find that the publication of these results in membranes constitutes a good choice since the peptides obtained are produced using bipolar membranes. This article will surely interest membrane readers by providing them new information about electro-membrane technologies that produce peptides with proven biological activities and where the innovative process using these membranes fits perfectly with the concept of circular economy.
